# High CD36 Expression Predicts Aggressive Invasion and Recurrence in Oral Squamous Cell Carcinoma

**DOI:** 10.3390/ijms262412071

**Published:** 2025-12-15

**Authors:** Kotaro Sakurai, Kei Tomihara, Manabu Yamazaki, Jun-ichi Tanuma, Shin-ichi Yamada

**Affiliations:** 1Department of Oral and Maxillofacial Surgery, Faculty of Medicine, University of Toyama, 2630 Sugitani, Toyama 930-0194, Japan; shinshin@med.u-toyama.ac.jp; 2Division of Oral and Maxillofacial Surgery, Faculty of Dentistry & Graduate School of Medical and Dental Sciences, Niigata University, 2-5274 Gakkocho-dori, Chuo-ku, Niigata 951-8514, Japan; tomihara@dent.niigata-u.ac.jp; 3Division of Oral Pathology, Faculty of Dentistry & Graduate School of Medical and Dental Sciences, Niigata University, 2-5274 Gakkocho-dori, Chuo-ku, Niigata 951-8514, Japan; manyamaz@dent.niigata-u.ac.jp (M.Y.); tanuma.dent@niigata-u.ac.jp (J.-i.T.)

**Keywords:** CD36, oral squamous cell carcinoma, recurrence

## Abstract

CD36, a fatty acid scavenger receptor expressed in tumors, is associated with a poor prognosis in several cancers. Our previous research demonstrated the involvement of CD36 in the proliferation and migration of oral squamous cell carcinoma (OSCC) cells. However, the clinical significance of CD36 expression in OSCC remains unclear. The purpose of this study was to evaluate the association between CD36 expression and the clinicopathological characteristics of OSCC patients. Immunohistochemical expression of CD36 was quantified using the H-score, and its association with clinicopathological characteristics was evaluated in 55 OSCC patients. The mean H-score for membrane-associated CD36 expression was 84.8. CD36 expression was significantly correlated with tumor stage, mode of invasion, differentiation, and recurrence of OSCC cells. Moreover, elevated CD36 expression was significantly correlated with a high rate of relapse. Univariate and multivariate analyses showed that CD36 expression was an independent risk factor for relapse. Moreover, The Cancer Genome Atlas (TCGA) dataset analysis revealed that CD36 expression may coexist with transcriptional activation of β-oxidation-related and epithelial–mesenchymal transition (EMT)-related pathways. These findings suggest that CD36 might serve as a predictive biomarker for OSCC malignancy and recurrence.

## 1. Introduction

Oral cancer, categorized as a type of head and neck cancer, ranks as the sixteenth most common malignancy worldwide [1]. Over 90% of oral cancers are squamous cell carcinomas [2]. Recent advancements in novel treatment strategies, such as molecular targeted therapy and immunotherapy, have shown significant promise in oral cancer treatment [3]. However, their clinical efficacy remains unsatisfactory in some patients because of drug resistance or acquired tolerance; therefore, the identification of alternative targetable molecules and the development of new treatment strategies are imperative.

CD36, a fatty acid membrane-associated receptor of approximately 80 kDa, binds to ligands such as fatty acids and lipoproteins [4]. CD36 is abundantly expressed in adipocytes, hepatocytes, and macrophages and is strongly associated with fat metabolism, taste perception, and the development of metabolic diseases such as diabetes and obesity [4]. Moreover, recent studies have revealed that the expression of CD36 in tumor cells may correlate with treatment outcomes and prognosis in various tumor types, including esophageal squamous cell carcinoma, lung squamous cell carcinoma, bladder cancer, luminal A breast cancer, and glioblastoma [5,6,7,8], suggesting that this molecule contributes to tumor progression.

Our previous study demonstrated that CD36 plays a role in the proliferation and migration of oral squamous cell carcinoma (OSCC), suggesting its potential as a novel therapeutic target for oral cancer [9]. However, the correlation between CD36 expression and the clinicopathological characteristics of OSCC remains unclear. Therefore, this study aimed to evaluate the association between CD36 expression and the clinicopathological characteristics of OSCC. Moreover, CD36 has been reported to be biologically associated with β-oxidation and epithelial–mesenchymal transition (EMT) [10,11]. Hence, we aimed to elucidate correlation with CD36 expression and genes related to fatty acid β-oxidation (*PPARA*, *ACADL*, *TXNIP*) and EMT (*ZEB1*, *ARG1*) using The Cancer Genome Atlas (TCGA) dataset analysis.

## 2. Results

### 2.1. Patients and Characteristics

The clinicopathological characteristics of patients are presented in Table 1. Of the 55 patients, 26 were male and 29 were female, with a median age of 73.0 years (range, 31–92 years). The median follow-up time for survivors was 22.0 months (7–152 months).

The primary sites included the tongue (*n* = 32, 58%), mandibular gingiva (*n* = 10, 18%), maxillary gingiva (*n* = 9, 16%), floor of the mouth (*n* = 2, 4%), buccal mucosa (*n* = 1, 2%), and lips (*n* = 1, 2%). Based on the Union for International Cancer Control (UICC) TNM classification criteria for oral cavity cancer, 8th edition [12], 22 (40%) patients were diagnosed with stage I, 17 (30%) with stage II, 8 (15%) with stage III, and 8 (15%) with stage IV. Histopathologically, OSCCs were classified as well differentiated (*n* = 26, 47%), moderately differentiated (*n* = 20, 36%), or poorly differentiated (*n* = 9, 16%), based on the World Health Organization classification [13]. The mode of invasion at the invasive front of the tumor was classified based on the criteria of Yamamoto et al. as grade 1 (*n* = 3, 5%), grade 2 (*n* = 16, 29%), grade 3 (*n* = 21, 38%), and grade 4C/4D (*n* = 15, 27%) [14]. Lymphovascular invasion was observed in 8 (15%), and perineural invasion in 3 (5%). Representative membrane-associated CD36 and H-E-stained photographs of the mode of invasion classification are shown in Figure 1. The mean H-scores for CD36 expression were 84.8 ± 46.9.

### 2.2. Correlations Between CD36 Expression and the Clinicopathological Characteristics of Patients with OSCC

The results from the analysis of the correlations between CD36 expression and the clinicopathological characteristics of patients with OSCC are presented in Table 1. High CD36 expression was significantly associated with the primary site (*p* = 0.0011), tumor stage (*p* = 0.026), differentiation (*p* = 0.013), mode of invasion (*p* = 0.0034), and recurrence (*p* = 0.0004).

### 2.3. Correlation Between CD36 Expression and Survival in Patients with OSCC After Treatment

The patient population was divided into two groups based on the median H-score (79.5) and analyzed according to CD36 expression, respectively. Overall survival (OS) rates showed no significant difference (*p* = 0.42, Figure 2). However, relapse-free survival (DFS) rates were significantly lower in patients with high CD36 expression than in those with low CD36 expression (*p* = 0.0002, Figure 3). Univariate analysis identified several significant prognostic factors for poor overall survival, including the T factor (*p* = 0.018), N factor (*p* = 0.048), tumor stage (*p* = 0.028), and mode of invasion (*p* = 0.04) (Table 2). CD36 expression was not relevant to overall survival. For relapse, significant prognostic factors included the T factor (*p* = 0.0014), N factor (*p* = 0.017), tumor stage (*p* = 0.016), mode of invasion (*p* < 0.0001), and CD36 expression (*p* = 0.0008) (Table 3). Moreover, multivariate analysis showed that the mode of invasion (*p* = 0.04, HR = 10.02, 95%CI = 1.48–197.7) was an independent risk factor for poor overall survival, whereas the mode of invasion (*p* = 0.04, HR = 2.62, 95%CI = 1.07–6.86) and CD36 expression (*p* = 0.036, HR = 3.29, 95%CI = 1.14–10.93) were independent risk factors for relapse (Table 2 and Table 3).

### 2.4. Correlation Between CD36 Expression and Related Molecular Profiles in Head and Neck Cancer from the TCGA Datasets

To further explore the molecular context of CD36, we analyzed RNA-seq data from 288 patients with head and neck cancer in TCGA Head and Neck Squamous Cell Carcinoma (HNSC) cohort. Genes related to fatty acid β-oxidation (*PPARA*, *ACADL*, *TXNIP*) and EMT (*ZEB1*, *ARG1*) were selected a priori because β-oxidation and EMT have been reported to be biologically associated with CD36 [10,15,16,17,18,19,20].

Spearman’s rank correlation analysis revealed modest but statistically significant positive associations between CD36 and several genes: *ACADL* (ρ = 0.29, *p* < 0.0001), *ARG1* (ρ = 0.41, *p* < 0.0001), *ZEB1* (ρ = 0.32, *p* < 0.0001), *PPARA* (ρ = 0.29, *p* < 0.0001), and *TXNIP* (ρ = 0.26, *p* < 0.0001) (Figure 4 and Figure 5).

These results suggest that CD36 expression may coexist with transcriptional activation of β-oxidation-related and EMT-related pathways, although the correlations were modest and do not imply causation.

## 3. Discussion

CD36 has been shown to play a significant role in various pro-tumor functions, including the regulation of proliferation, metastasis, resistance to chemotherapy or radiotherapy, and angiogenesis [21,22]. Moreover, recent clinical studies have shown a correlation between high CD36 expression and a poor prognosis in several cancers, including esophageal squamous cell carcinoma, gastric cancer, lung squamous cell carcinoma, bladder cancer, luminal A breast cancer, glioblastoma, and acute myeloid leukemia [5,6,7,8,21,23]. However, the correlation between CD36 expression and the clinicopathological characteristics of oral cancer remains unclear. The purpose of this study was to examine the clinicopathological role of CD36 in OSCC. Consequently, high CD36 expression was significantly correlated with high malignancy in oral cancer.

Our previous study highlighted the biological activity of CD36, including its involvement in the proliferation and migration of OSCC cells [9]. More recently, we demonstrated that selective inhibition of CD36 can induce antitumor immunomodulatory effects in a mouse model of oral cancer [24]. Furthermore, the regulatory function of CD36 in EMT has been demonstrated in hepatocellular carcinoma and cervical cancer cells [11,20]. Taken together, in OSCC, CD36 is correlated to high malignancy and may be associated with EMT and become a potential diagnostic biomarker or therapeutic target.

In this clinicopathological study, high levels of CD36 immunohistochemical expression were significantly associated with poor histological differentiation and high-grade 4C/4D mode of invasion. Moreover, patients with high levels of CD36 expression showed a high rate of recurrence. Multivariate analysis revealed high CD36 expression and high-grade 4C/4D mode of invasion as a significant prognostic factor for relapse. A significant difference in relapse-free survival was observed, but not in overall survival. This may be explained by the fact that most patients had early-stage cancer, few patients relapsed and subsequently died after additional treatment, and the follow-up period was relatively short.

The possible mechanism by which CD36 affects recurrence is the cancellation of dormancy for cancer cells due to oxidative stress. Increased fatty acid storage due to high CD36 expression promotes β-oxidation, which results in increased reactive oxygen species (ROS) and oxidative stress [25]. Oxidative stress is a candidate factor for recurrence of dormant cancer cells [26]. High expression of CD36 may be indicator of recurrence in OSCC. For instance, in papillary thyroid cancer, high expression of CD36 on macrophages increases the recurrence rate [27]. It has also been reported that high expression of CD36 contributes to an increased recurrence rate in pancreatic ductal adenocarcinoma and acute myeloid leukemia [23,28]. On the other hand, in this study, high-grade 4C/4D mode of invasion was also associated with recurrence. Previous studies have shown that the high-grade 4C/4D mode of tumor invasion is associated with high malignancy and metastasis [14,29]. Phenotypic alterations like cell scattering occur via EMT; therefore, the mode of invasion may be related to EMT [30].

Through EMT, epithelial markers such as E-cadherin are downregulated, while mesenchymal markers and cellular motility are enhanced, leading to increased local invasion and changes in the pattern of infiltration, such as cord-like or diffuse invasion corresponding to grade 4C/4D mode of invasion [31,32]. These features are considered factors contributing to an increased risk of local recurrence or metastasis after surgical resection. However, EMT and histological dedifferentiation are distinct concepts. This is consistent with the results of the multivariate analysis of this study.

To further understand the molecular context of CD36, we performed correlation analyses using TCGA data. Interestingly, this analysis revealed that *PPARA*, *ACADL*, *TXNIP*, *ZEB1*, and *ARG1* showed a significant positive correlation with CD36 expression. *PPARA* and *ACADL* are related to β-oxidation [15,16]. *TXNIP* is involved in the production of ROS [17]. These molecules are significantly correlated with CD36, and high expression of CD36 may increase β-oxidation and ROS production, which may contribute to the recurrence of dormant cancer cells. Moreover, amplification of *ZEB1* or *ARG1* is relevant to EMT or metastasis [18,19]. Correlation analysis suggests that EMT-related molecules such as *ZEB1* and *ARG1* may be co-expressed with CD36. These findings indicate that high CD36 expression is associated with EMT. Additionally, high CD36 expression may be correlated to poor histological differentiation and an aggressive invasion pattern (grade 4C/4D mode of invasion). Further studies are required to clarify the underlying causal mechanisms.

This study has several limitations. First, the sample size was relatively small and restricted to surgically resectable cases, which may limit the generalizability of the findings. Second, the TCGA analysis was based solely on mRNA expression and did not include protein-level validation. Although CD36 expression showed positive correlations with EMT-related genes such as *ZEB1* and *ARG1*, these results only indicate possible transcriptional co-expression and do not establish a causal relationship. Therefore, functional studies are required to clarify whether CD36 directly regulates EMT or invasion pathways in OSCC.

In conclusion, our results demonstrated that CD36 is associated with OSCC malignancy and may serve as a novel diagnostic biomarker. However, no association with overall survival was observed, suggesting that CD36 may have limited significance as a prognostic predictor. The possibility that CD36 expression is involved in the mechanism of recurrence is clinically important and provides evidence that will serve as a basis for future functional studies and consideration as a therapeutic target.

## 4. Materials and Methods

### 4.1. Patients

This study included 55 patients who underwent surgical treatment for OSCC at the Department of Oral and Maxillofacial Surgery, University of Toyama Hospital, between April 2009 and March 2024. The inclusion criterion was a cytological or histopathological diagnosis of OSCC, and inclusion was also limited by sample availability. No predefined exclusion criteria were applied. Nevertheless, none of the tissue specimens showed positive surgical margins, and no patients in our cohort had received preoperative chemotherapy or radiotherapy or had immunosuppression or other conditions likely to substantially affect survival or molecular profiles. Medical records were retrospectively examined for age, sex, body mass index (BMI), primary site, tumor stage, T factor, N factor, differentiation, mode of invasion, lymphovascular invasion, perineural invasion and recurrence. The tumor extent and the histopathological grading were classified based on the UICC TNM classification criteria for oral cavity cancer, 8th edition [12]. The mode of invasion at the invasive front of the tumor was classified according to Yamamoto et al. [14]. This study was conducted in accordance with the ethical guidelines outlined in the Declaration of Helsinki and was approved by Research Ethics Committee for Clinical and Epidemiological Research of Toyama University (Approval No. R2018168; Approval Date 19 April 2019). Owing to the retrospective, observational nature of this study, informed consent was obtained using an opt-in/opt-out approach.

### 4.2. Immunohistochemistry

Immunohistochemical analysis of paraffin-embedded samples was performed. The thickness of the sections was 4 μm. For antigen retrieval, the sections were immersed in citric acid buffer (pH 6.0) and heated at 121 °C for 10 min. The deparaffinized and rehydrated sections were immersed in 0.3% hydrogen peroxide in methanol for 10 min to block endogenous peroxidase activity. To prevent nonspecific binding of the antibodies, the sections were immersed in a blocking solution (Nacalai Tesque, Kyoto, Japan) for 15 min. Subsequently, the sections were incubated with the primary antibody (anti-CD36 rabbit monoclonal antibody, 1:100; clone EPR6573; Abcam, Cambridge, UK) overnight at 4 °C. After incubation, the sections were immersed in phosphoric acid thrice for 3 min. Peroxidase reactions were developed using a 3,3’-diaminobenzidine tetrahydrochloride substrate solution (DAKO, Glostrup, Denmark), and the sections were counterstained with hematoxylin. The same primary antibody was used as a positive control.

### 4.3. Quantification of CD36 Expression with Immunoreactivity Scoring

CD36 expression was quantified based on H-score [33]. Immunohistochemically stained specimens were captured to create virtual slides, and immunoreactivity was analyzed using Aperio analysis software (Leica Biosystems, Tokyo, Japan) at ×20 magnification with objective lens. Expression intensity was categorized into four stages: no expression, 0; weak, 1+; moderate, 2+; and strong, 3+. The distribution of expression was presented as a percentage of the number of cells (numerator) relative to the total number of cells in the analysis area (denominator). The H-score was calculated using the following formula: H-score (100−300) = (% staining area of “1+”) × 1 + (% staining area of “2+”) × 2 + (% staining area of “3+”) × 3.

### 4.4. Statistical Analysis

Comparisons were made using the Mann–Whitney U test or Kruskal–Wallis test for continuous variables and the chi-square test or Fisher’s exact test for categorical variables. Correlations were analyzed using Spearman’s rank correlation. Survival curves were generated using the Kaplan–Meier method, and differences between overall survival curves were evaluated using the log-rank test. Univariate and multivariate analyses were performed using Cox’s proportional hazards model. Differences were considered statistically significant at *p* < 0.05. All statistical analyses were performed using GraphPad Prism software v.9 (GraphPad Software, Boston, MA, USA).

### 4.5. Acquisition and Analysis of TCGA Data

A total of 288 RNA-seq data from the TCGA-HNSC cohort were obtained via the Genomic Data Commons (GDC) portal [34]. TCGA data were accessed on October 14, 2025. As a breakdown, oral cavity, oropharyngeal, laryngeal/hypopharyngeal and unknown samples included 172, 16, 59, and 41 cases, respectively. Gene-level expression (STAR-counts workflow) was normalized, log2-transformed, and analyzed using R software. Spearman’s rank correlation coefficients (ρ) were calculated to assess monotonic relationships between CD36 and candidate genes related to fatty acid β-oxidation (*PPARA*, *ACADL*, *TXNIP*) or EMT/metastasis (*ZEB1*, *ARG1*). Scatter plots were visualized using LOESS smoothing with 95% confidence intervals to illustrate overall expression trends.

## Figures and Tables

**Figure 1 ijms-26-12071-f001:**
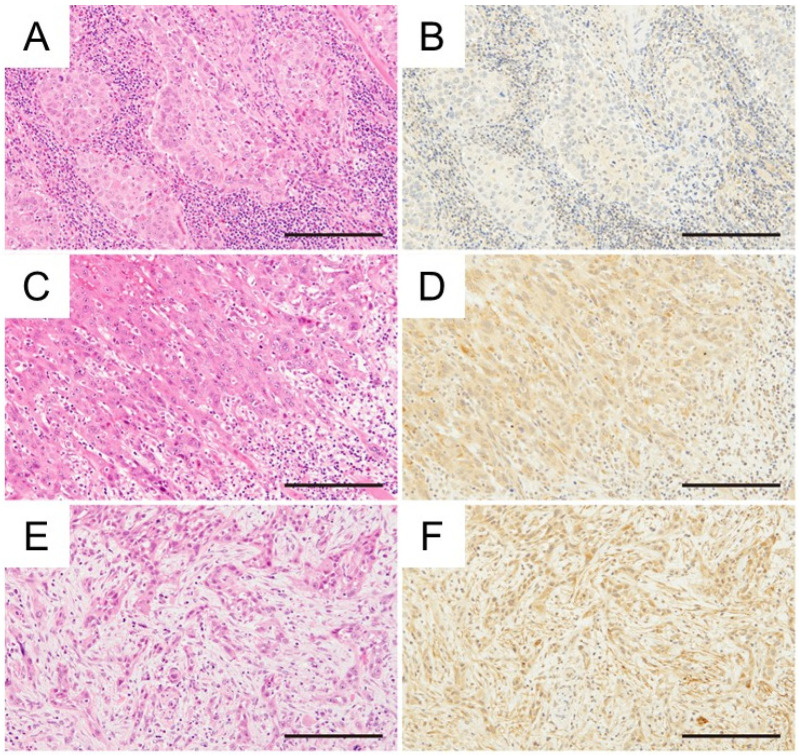
Immunohistochemical investigation between the mode of invasion and CD36 expression in OSCC (H&E staining, immunohistochemical CD36 staining) (**A**,**B**) Grade 3, tongue cancer, H-score: 77.0. (**C**,**D**) Grade 4C, tongue cancer, H-score: 125.6. (**E**,**F**) Grade 4D, tongue cancer, H-score: 135.2. Magnification ×400, scale bar: 200 μm.

**Figure 2 ijms-26-12071-f002:**
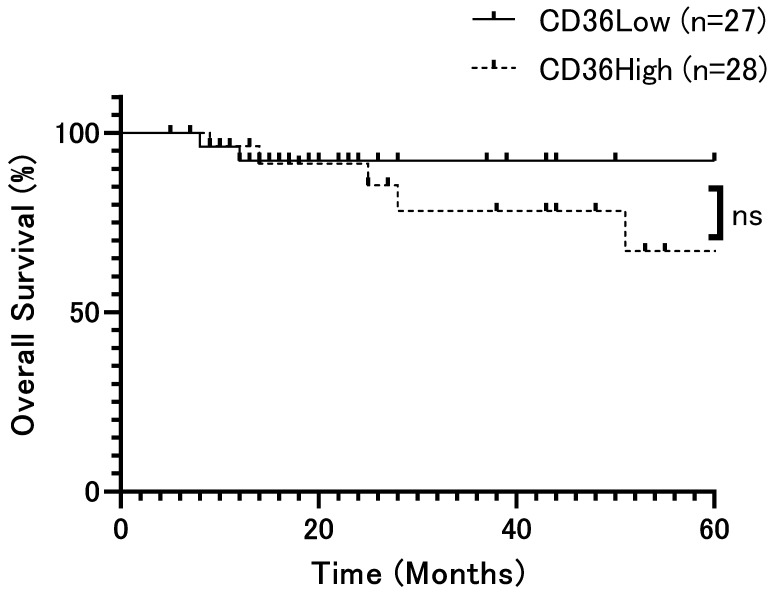
Overall survival rate of patients with OSCC according to CD36 expression (*p* = 0.42). Overall survival rates showed no significant difference. The x-axis represents time (months), and the y-axis represents overall survival (%). ns: not significant.

**Figure 3 ijms-26-12071-f003:**
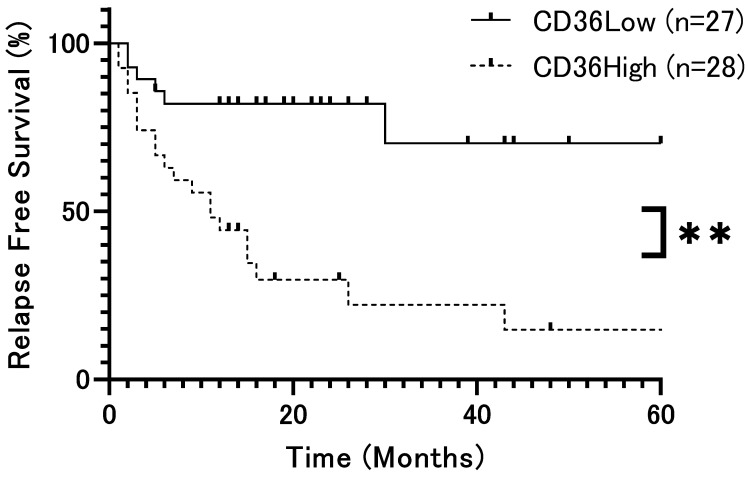
Relapse-free survival rate of patients with OSCC according to CD36 expression (*p* = 0.0002). Relapse-free survival rates were significantly lower in patients with high CD36 expression than in those with low CD36 expression. The x-axis represents time (months), and the y-axis represents relapse-free survival (%). **: *p* < 0.01.

**Figure 4 ijms-26-12071-f004:**
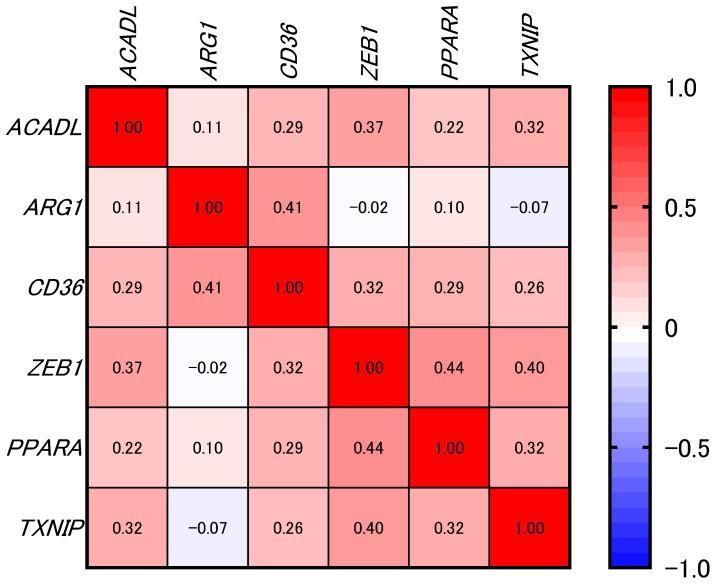
Heatmap showing the correlations between CD36 and selected genes of interest (not filtered by significance) in samples from the TCGA-HNSC dataset (*n* = 288). The numbers inside the cells of the heatmap represent correlation coefficients. The color key on the right shows the range of Spearman’s rank correlation coefficients (ρ), with red indicating values close to 1 (positive correlations) and blue indicating values close to −1 (negative correlations). Correlation was calculated using Spearman’s rank correlation test.

**Figure 5 ijms-26-12071-f005:**
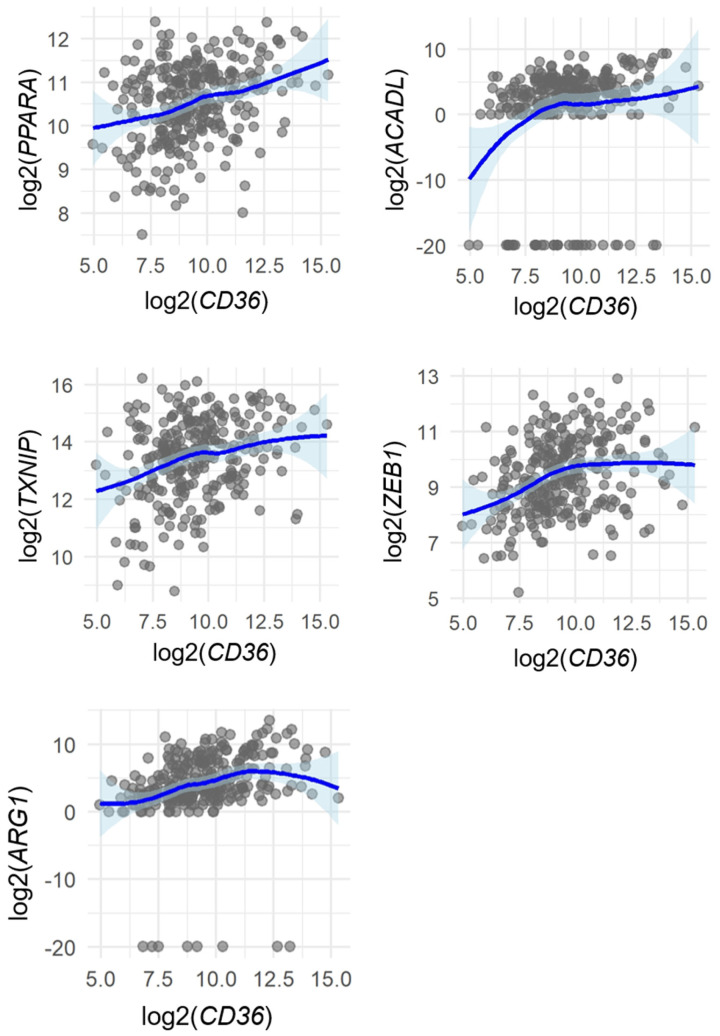
Spearman’s rank correlations between CD36 and selected genes (*PPARA*, *ACADL*, *TXNIP*, *ZEB1*, and *ARG1*) in samples from the TCGA-HNSC dataset (*n* = 288). Each dot represents one sample. LOESS (locally weighted regression) curves with 95% confidence intervals are shown to visualize overall trends without assuming linearity. Correlation strength and *p*-values were determined using Spearman’s rank correlation coefficient (ρ).

**Table 1 ijms-26-12071-t001:** Correlation between CD36 expression and clinicopathological characteristics in patients with OSCC.

	N (%)	Median (Interquartile Range)	*p*-Values
**Age, years**			0.45
<72	27 (49%)	97.4 (3.5–165.8)	
>72	28 (51%)	66.7 (1.3–156.0)	
**Sex**			0.21
Male	26 (47%)	104.7 (1.3–165.8)	
Female	29 (53%)	64.7 (4.2–156.0)	
**BMI**			0.37
<21.7	27 (49%)	64.9 (4.2–165.8)	
>21.7	28 (51%)	98.0 (1.3–155.3)	
**Primary site**			0.0011
Tongue	32 (58%)	108.7 (5.9–165.8)	
Mandibular gingiva	10 (18%)	12.2 (1.3–148.9)	
Maxillary gingiva	9 (16%)	63.3 (35.3–79.5)	
Floor of mouth	2 (4%)	155.7 (155.3–156.0)	
Buccal mucosa	1 (2%)	129.2	
Lip	1 (2%)	54.4	
**Tumor stage**			0.026
I	22 (40%)	54.2 (1.3–145.4)	
II	17 (31%)	98.5 (42.7–155.3)	
III	8 (15%)	65.2 (4.2–140.1)	
IV	8 (15%)	118.6 (64.7–165.8)	
**T factor**			0.065
T1	22 (40%)	54.2 (1.3–145.4)	
T2	21 (38%)	98.5 (4.2–155.3)	
T3	6 (11%)	89.6 (53.9–138.6)	
T4	6 (11%)	103.1 (64.7–165.8)	
**N factor**			0.25
N0	44 (80%)	74.0 (1.3–156.0)	
N1–3	11 (20%)	125.7 (4.2–165.8)	
**Differentiation**			0.013
Well	25 (45%)	64.7 (1.3–145.4)	
Moderate	21 (38%)	74.1 (3.5–156.0)	
Poorly	9 (16%)	125.7 (61.6–165.8)	
**Mode of invasion**			0.0034
1	3 (5%)	58.7 (52.5–127.9)	
2	16 (29%)	53.9 (1.3–145.4)	
3	21 (38%)	79.5 (3.5–156.0)	
4C/4D	15 (27%)	125.7 (60.3–165.8)	
**Lymphovascular invasion**			0.13
Absent	47 (85%)	68.7 (1.3–165.8)	
Present	8 (15%)	104.5 (66.1–145.6)	
**Perineural invasion**			0.15
Absent	52 (95%)	74.0 (1.3–165.8)	
Present	3 (5%)	121.7 (110.5–145.6)	
**Recurrence**			0.0004
Absent	29 (53%)	58.7 (1.3–140.1)	
Present	26 (47%)	125.7 (4.2–165.8)	

**Table 2 ijms-26-12071-t002:** Results of univariate and multivariate analyses of clinicopathological factors affecting overall survival rates following surgery.

Characteristics	Univariate Analysis			Multivariate Analysis		
	HR	95%CI	*p* Value	HR	95%CI	*p* value
Age (years)						
<72 versus >72	0.84	0.16 ~ 3.85	0.82			
Sex						
Female versus male	1.53	0.34 ~ 7.76	0.58			
BMI						
<21.7 versus >21.7	1.18	0.25 ~ 6.03	0.83			
T factor						
T1–2 versus T3–4	6.12	1.34 ~ 31.27	0.018	1.29	0.14 ~ 27.59	0.83
N factor						
Absent versus present	4.58	0.99 ~ 23.5	0.048	0.27	0.0082 ~ 8.27	0.4
Tumor stage						
I-II versus III-IV	6.31	1.35 ~ 44.13	0.028	19.99	0.24 ~ 1728	0.16
Differentiation						
Well versus moderate–poorly	2.06	0.44 ~ 14.46	0.39			
Mode of invasion						
1–3 versus 4C/4D	5.78	1.18 ~ 41.37	0.04	10.02	1.48 ~ 197.7	0.04
Lymphovascular invasion						
Absent versus present	0.96	0.051 ~ 5.63	0.97			
Perineural invasion						
Absent versus present	1.3 × 10^−11^	-	>0.99			
CD36 expression						
Low versus high	1.93	0.37 ~ 13.94	0.45			
Recurrence						
Absent versus present	2.5 × 10^−12^	3.88 ~ NE	>0.99			

HR hazard risk, CI confidence interval, NE not estimable due to model instability caused by sparse events.

**Table 3 ijms-26-12071-t003:** Results of univariate and multivariate analyses of clinicopathological factors affecting relapse-free survival rates following surgery.

Characteristics	Univariate Analysis			Multivariate Analysis		
	HR	95%CI	*p* Value	HR	95%CI	*p* Value
Age (years)						
<72 versus >72	0.92	0.041 ~ 2.00	0.83			
Sex						
Female versus male	0.73	0.33 ~ 1.59	0.44			
BMI						
<21.7 versus >21.7	1.75	0.8 ~ 3.99	0.17			
T factor						
T1–2 versus T3–4	4.009	1.65 ~ 9.27	0.0014	2.47	0.54 ~ 17.75	0.29
N factor						
Absent versus present	2.7	1.15 ~ 5.96	0.017	0.59	0.11 ~ 4.62	0.56
Tumor stage						
I-II versus III-IV	2.62	1.17 ~ 5.71	0.016	2.1	0.15 ~ 20.17	0.54
Differentiation						
Well versus moderate–poorly	1.57	0.72 ~ 3.52	0.26			
Mode of invasion						
1–3 versus 4C/4D	4.94	2.26 ~ 10.97	<0.0001	2.62	1.07 ~ 6.86	0.04
Lymphovascular invasion						
Absent versus present	2.33	0.9 ~ 5.34	0.058			
Perineural invasion						
Absent versus present	0.55	0.031 ~ 2.59	0.56			
CD36 expression						
Low versus high	4.85	2.05 ~ 13.34	0.0008	3.29	1.14 ~ 10.93	0.036

HR hazard risk, CI confidence interval.

## Data Availability

The dataset is available at https://doi.org/10.6084/m9.figshare.28254308.v1.

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
