# Peer review of "High CD36 Expression Predicts Aggressive Invasion and Recurrence in Oral Squamous Cell Carcinoma"

_ijms, 2025, doi:10.3390/ijms262412071_

Round 1

Reviewer 1 Report

Comments and Suggestions for Authors

In the presented manuscript, the Authors examined the association between CD36 expression and the clinicopathological characteristics of OSCC patients.
I enjoyed reading this article. However, I have a few comments/suggestions.

  1. I recommend adding study inclusion criteria and exclusion criteria (Chapter 4.1. Patients).
  2. I recommend adding a description of the data obtained from the TCGA-HNSC. How many samples were analysed (this information can only be found in the figure in the Results section)? When was the database accessed? The expansion of the TCGA abbreviation is also missing.
  3. Gene symbols should be italicised.

Reviewer 2 Report

Comments and Suggestions for Authors

The study „High CD36 expression predicts aggressive invasion and recurrence in oral squamous cell carcinoma” by Kotaro Sakurai et al., analyzed the association between CD36 expression and the clinicopathological characteristics of OSCC patients. The Authors suggested that CD36 might serve as a predictive biomarker for OSCC malignancy and recurrence.

The manuscript requires some corrections

In the introduction, the authors omit the fact that analyses were also performed using TCGA datasets. It's worth mentioning this and explaining why these genes were chosen for analysis, especially since they write in line 124” ...were selected a priori based on their reported biological relevance to CD36.” Moreover, the gene names should be written in italics.

Discussion: The Authors write that :” High expression of CD36 may be indicator of recurrence. [line 175].  It is worth comparing this observation with other studies on oncological patients with HNSCC or other cancers.

In the part of Material and methods :

4.1. The authors did not provide criteria for inclusion and exclusion of patients from the study.

4.5. it is worth adding how many samples were analyzed in total from the TCGA database, as well as how many samples were in the mentioned locations.

References:

The reference to the WHO classification  [line 66] should probably be different from 9, which referes to the UICC.

Is there a missing reference in line 35? Moreover, the reference from line 39 is not necessary if you are still quoting from the same source.

Table and Figures

In table 2 - I suggest using the power of ten in lines of „Perineural invasion” and „Recurrence”. Similar in this same line, it would be better to use an en dash or em dash instead of a question mark.

In figures 2, 3, and 4 the axis description is illegible.

The abbreviation should be explained on first use

UICC [line 61]

TCGA-HNSC [line 122]  - on the same line there is no unnecessary explanation of the abbreviation OSCC

RFS and OS [line 168]

Round 2

Reviewer 1 Report

Comments and Suggestions for Authors

The Authors have corrected the article in response to all my comments and suggestions.

Reviewer 2 Report

Comments and Suggestions for Authors

The Authors made corrections to the manuscript.